# Rotation Control Unlearning: Quantifying and Controlling Continuous Unlearning for LLM with The Cognitive Rotation Space

## Abstract

As Large Language Models (LLMs) become increasingly prevalent, their security vulnerabilities have already drawn attention. Machine unlearning is introduced to seek to mitigate these risks by removing the influence of undesirable data. However, existing methods not only rely on the retained dataset to preserve model utility, but also suffer from cumulative catastrophic utility loss under continuous unlearning requests. To solve this dilemma, we propose a novel method, called Rotation Control Unlearning (RCU), which leverages the rotational salience weight of RCU to quantify and control the unlearning degree in the continuous unlearning process. The skew symmetric loss is designed to construct the existence of the cognitive rotation space, where the changes of rotational angle can simulate the continuous unlearning process. Furthermore, we design an orthogonal rotation axes regularization to enforce mutually perpendicular rotation directions for continuous unlearning requests, effectively minimizing interference and addressing cumulative catastrophic utility loss. Experiments on multiple datasets confirm that our method without retained dataset achieves SOTA performance.

## 1 Introduction

In recent years, the development of Large Language Models (LLMs) has received widespread attention. With the extensive application of GPT (Achiam et al. (2023)) and other LLMs (Liu et al. (2024); Touvron et al. (2023)) in academic research and industry (Wang et al. (2024)), concerns about LLMs have also increased. Among these, security issues regarding information protection have become particularly prominent. These concerns have motivated researchers to use the machine unlearning method to remove potentially private (Ortiz-Jimenez et al. (2023)), illegal or toxic data that may exist in LLMs. Currently, machine unlearning in LLMs (Bourtoule et al. (2021)) is mainly divided into two paradigms: the method based on parameters (Chen & Yang (2023b); Eldan & Russinovich (2023); Jia et al. (2024)) and the method based on in-context unlearning (Thaker et al. (2024); Pan et al. (2020)). The methods based on parameters achieve effective unlearning by maximizing the task loss on the unlearning data (Wang et al. (2025a); Hu et al. (2025)). The methods based on in-context unlearning modify the input prompts of LLM to make them refuse to output the content that needs to be unlearning (Chen et al. (2025); Yu et al. (2025)). Other methods achieve the unlearning goal by interfering with the LLM's representation of the unlearned data (He et al. (2025); Jiang et al. (2025)).

However, unlearning methods in LLM are often not a one-time operation but a continuous process in real world. Most of them exist the cumulative catastrophic utility loss (Gao et al. (2025)) when dealing with continuous unlearning. The cumulative catastrophic utility loss causes a significant decline in both the LLM's unlearning capability and utility retention capacity during the continuous unlearning process as the number of requests increases. At the parameter level, this manifests as new unlearning requests inducing parameter shift in previously learned ones. Furthermore, they still require a retained dataset to maintain the model's utility. This retained dataset consists of a part of the original training dataset (Bourtoule et al. (2021)). Since LLM require a large amount of data for training (Wang et al. (2024)), using the retained dataset in continuous unlearning is not feasible (Liu et al. (2025)).

The work of $o^3$ (Gao et al. (2025)) proposes to mitigate cumulative catastrophic utility loss by imposing orthogonal constraints on LoRA parameters and introduces weights for the LoRA (Hu et al. (2022)) modules to represent the degree of unlearning. However, this approach suffers from several significant limitations. Firstly, the effectiveness of its simple orthogonal constraints on LoRA parameters diminishes as the number of unlearning requests increases, making it difficult to sustainably alleviate cumulative catastrophic utility loss. Secondly, using LoRA weights to quantify the degree of unlearning lacks interpretable justification. Finally, the mapping from the Out-Of-Distribution (OOD) detector outputs to the corresponding weights heavily relies on empirical design, which substantially increases the complexity and cost of application.

In this work, we propose Rotation Control Unlearning (RCU), a novel unlearning method that addresses the above challenges. This method is inspired by the theory of Lie group (Gallier (2001)) and re-constructs the LoRA update paradigm through mathematical derivation. It re-expresses the unlearning update of LLM as rotational operations within a cognitive rotation space. The cognitive rotation space is defined as a high-dimensional rotation space, which is used to depict the rotational transformations experienced by the original parameters of LLM during continuous unlearning. This enables the transformation of the uncontrollable parameter shift into controllable rotational angle changes, thereby effectively alleviating the cumulative catastrophic utility loss. The specific mathematical formulation is elaborated in the methodology. Our approach introduces a skew symmetric loss in LoRA update paradigm to formulate the unlearning process as rotation operations, with the rotational angle serving as a precise quantification metric. We introduce an orthogonal rotation axes loss to enforce perpendicular rotation directions for consecutive unlearning requests, effectively mitigating cumulative catastrophic utility loss by minimizing inter unlearning request interference. Furthermore, to enhance compatibility, we design an unlearning alignment loss that guides the OOD detector to produce representations aligned with our LoRA update paradigm. These representations then collaborate with the distributional shift compensator to generate rotational salience weights for auxiliary quantification. Finally, our method is supported by straightforward experimental interpretability and requires significantly fewer trainable parameters than $o^3$.

Specifically, our contributions are outlined as follows:

- We propose the RCU method, which quantifies the unlearning process by leveraging rotational changes in the cognitive rotation space, and introduce the rotational salience weight to precisely control the degree of unlearning throughout the continuous unlearning process.

- We design the skew symmetric loss to establish the existence of the cognitive rotation space and the orthogonal rotation axes loss to alleviate cumulative catastrophic utility loss.

- We demonstrate the connection between rotation and unlearning through mathematical proof and experimental validation.

- Extensive experiments on the ScienceQA and TOFU datasets confirm the effectiveness of our proposed method without retained dataset.

## 2 PRELIMINARY

**Machine Unlearning in LLMs.** The objective of machine unlearning is to safeguard information security. Currently, there are two mainstream approaches: parameters-based methods Chen & Yang (2023b); Eldan & Russinovich (2023); Jia et al. (2024) and in-context unlearning-based methods. Thaker et al. (2024); Pan et al. (2020) Parameters-based methods iteratively adjust the LLMs' internal parameters to minimize the loss function on specific tasks, thereby improving unlearning performance (Yi et al. (2025); Yang et al. (2025); Bronec & Helcl (2025); Premptis et al. (2025)). Yi et al. (2025) method employs fine-tuning for rapid learning and induces deliberate model degradation upon detection of harmful fine-tuning behaviors. Yang et al. (2025) approach utilizes a reweighting strategy to adjust training sample weights, focusing particularly on data useful for unlearning. In-context unlearning-based methods modify input prompts to prevent the generation of undesired content. Yu et al. (2025) method generates tokens that guide forgetting based on the input query, achieving unlearning without altering model parameters. Additionally, other techniques exist (Muhamed et al. (2025); Wang et al. (2025b)). For instance, Muhamed et al. (2025)s control forgetting by manipulating model activations. The He et al. (2025) method disrupts the latent space of forgotten samples during training to induce chaotic outputs. The Zhao et al. (2025) has achieved

ideal unlearning by masking out the training signal of TOFU dataset (Maini et al. (2024)) in their corpus from the data perspective. While existing methods often overlook the challenges of continuous unlearning requests and the associated catastrophic degradation of model utility in real-world scenarios, Gao et al. (2025) formalizes the concept of continuous machine unlearning and introduces an unlearning framework based on an out-of-distribution detector. Building upon the Gao et al. (2025) paradigm, our method proposes a more refined LoRA update strategy that enables more precise quantification of unlearning extent.

**Out-Of-Distribution Detection.** The current methods of OOD detection include one-class SVM based methods Erfani et al. (2016), random forest based methods Mihaylov et al. (2018), Gaussian mixture modeling based methods Laxhammar et al. (2009), and deep learning based OOD detection methods Yang et al. (2024). At the same time, OOD detection based on deep learning has become the mainstream in classification tasks. Among them, Zong et al. (2018) is a method suitable for multi-source time series, which estimates OOD scores by generating low-dimensional representations through deep autoencoders. Xu et al. (2021) extracts features by pre-trained language model and then fits one-class SVM for detection. In addition, Zhou et al. (2023) using ensemble learning, Lang et al. (2022) using pseudo-label, Cao et al. (2024) using outlier exposure, Ouyang et al. (2023) using prefix adjustment and other methods have achieved good results in OOD detection. Gao et al. (2025) incorporates the contrastive entropy loss and Masked Language Modeling (MLM) loss (Jian et al. (2022)), enhances the ability to detect out-of-distribution cases. By studying the unlearning process, we introduced the unlearning alignment loss, thereby enhancing the compatibility between the OOD detector and unlearning.

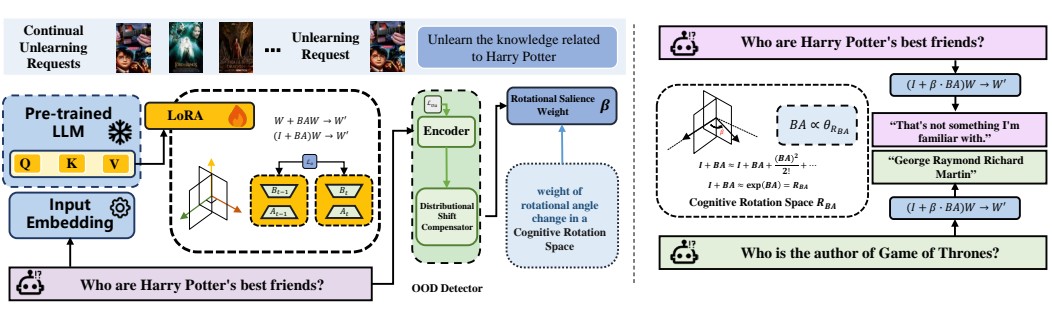

Figure 1: The overall architecture of our method is shown in the figure. In the training pipeline, the orthogonal rotation axes loss $\mathcal{L}_o$ is applied to the attention layers of the LLMs for training; simultaneously, the unlearning alignment loss $\mathcal{L}_{Ua}$ is used to train an OOD detector, whose output is fed into the distributional shift compensator to generate the rotational salience weight $\beta$. In the inference pipeline, given that the LoRA parameters $BA$ are proportional to the rotation angle $\theta_{R_{BA}}$ in the Cognitive Rotation Space $R_{BA}$. We control the rotation angle $\theta_{R_{BA}}$ amplitude by adjusting the scale of LoRA $BA$, and use the weight $\beta$ to dynamically load the parameters that match the required unlearning degree.

## 3 METHODOLOGY

**Problem Definition.** We use the popular causal LLMs, where the input to the LLM is a sequence of text tokens of variable length. The LLM $M_\Theta$, where $\Theta$ is the parameter of the LLM, will calculate the probability of each token in the text under the preorder token based on the input. We set continuous unlearning ploblem as a series of consecutive arriving unlearning requests, each with $N^{U,t}$ data samples, which can be written as $\left\{ D^{U,t} \right\}_{t=1}^{T}$. For the $t - th$ unlearning request , $D^{U,t} = \left\{ x_i || x_i \sim \mathcal{P}_{\mathcal{X}}^{U,t} \right\}_{i=1}^{N^{U,t}}$ , where $T$ is the index of the latest arriving unlearning request, and the $P$ is the input marginal distribution. In each request, we utilize the input $\mathcal{P}_{\mathcal{X}}^t$ and the label distribution $\mathcal{P}_{\mathcal{Y}}^t$ for training. Traditional unlearning methods assume a holdout data set drawn from a distribution $\mathcal{P}_{\mathcal{Y}}^{R,t}$ that is disjoint from the forgetting data set $\mathcal{P}_{\mathcal{X}}^{U,t}$ to preserve the performance of the

model on the original training distribution. The immediate goal of continuous unlearning is:

$$\sum_{t=1}^{T} \min_{\Theta^t} \mathbf{I}\left(M_{\boldsymbol{x}\sim\mathcal{P}_{\mathcal{X}}^{\mathrm{U},t}}(\boldsymbol{x},\Theta); M^t_{\boldsymbol{x}\sim\mathcal{P}_{\mathcal{X}}^{\mathrm{U},t}}\left(\boldsymbol{x},\Theta^t\right)\right), \sum_{t=1}^{T} \max_{\Theta^t} \mathbf{I}\left(M_{\boldsymbol{x}\sim\mathcal{P}_{\mathcal{X}}^{\mathrm{R},t}}(\boldsymbol{x},\Theta); M^t_{\boldsymbol{x}\sim\mathcal{P}_{\mathcal{X}}^{\mathrm{R},t}}\left(\boldsymbol{x},\Theta^t\right)\right), \tag{1}$$

where $M^t$ with the parameters $\Theta^t$ represents the target model during and after unlearning on the $t-th$ unlearning set $D^{U,t}$, and $I\left(\cdot;\cdot\right)$ computes the mutual information between two random variables. The model utility preservation on other distributions $\mathcal{P}_{\mathcal{X}}^{o}$ different from the unlearning distribution is another goal of unlearning. This can be expressed as follows:

$$\sum_{t=1}^{T} \max_{\Theta^t} \mathbf{I}\left(M_{\boldsymbol{x}\sim\mathcal{P}_{\mathcal{X}}^{\mathcal{O}}}(\boldsymbol{x},\Theta); M^t_{\boldsymbol{x}\sim\mathcal{P}_{\mathcal{X}}^{\mathcal{O}}}\left(\boldsymbol{x},\Theta^t\right)\right). \tag{2}$$

### 3.1 CONTINUOUS UNLEARNING FOR LLM WITH LoRA

The continuous unlearning process of LLM inevitably leads to cumulative catastrophic utility loss (Gao et al. (2025)). The cumulative catastrophic utility loss manifests as a significant decline in both the LLM's unlearning capability and its utility retention capability as the number of unlearning requests increases. At the parameter level, this is reflected in a shift of the parameters corresponding to previous unlearning requests when the model is trained on new ones. This requires the method to simultaneously achieve the continuous preservation of its historical unlearning knowledge and the original utility when handling the current unlearning request.

While existing approaches $o^3$ (Gao et al. (2025)) rely on orthogonal constraints to enforce perpendicularity between parameters of $A$ ($W = BA$). However, this constraint suffers from inherent limitations: the introduction of parameter $B$ undermines its effectiveness, and its capability further diminishes as the number of unlearning requests accumulates, making it inadequate for continuous unlearning scenarios. Moreover, the $o^3$ empirically assigns weights to LoRA parameters to represent the degree of unlearning, which is a heuristic design that lacks theoretical grounding and leads to poor interpretability.

We proposed the RCU to address these challenges. We develop a mathematically-derived approach with enhanced interpretability for quantifying unlearning process. By constructing a cognitive rotation space wherein LLM parameter updates are formulated as a novel rotational transformation, the RCU transforms continuous unlearning into rotations in a high-dimensional parameter space, thereby converting uncontrollable parameter shift into controlled angular rotations and ultimately alleviating cumulative catastrophic utility loss.

Our analysis in Figure 2 (a)(b) revealed the relationship between different weighting coefficients $\beta$ and unlearning updates, demonstrating that the extent of unlearning intensifies with increasing values of the $\beta$.

LoRA Hu et al. (2022) reduces the trainable parameters by introducing two low-rank trainable matrices $\{A, B\}$, where $W_{LoRA} = BA, W_{LoRA} \in \mathbb{R}^{U \times V}, B \in \mathbb{R}^{U \times K}, A \in \mathbb{R}^{K \times V}$, which decomposes the high-dimensional matrix into a low-rank matrix. The specific update formula is as follows:

$$W' \leftarrow W + BA. \tag{3}$$

LoRA updates inevitably lead to uncontrollable parameter shift. To address this problem, we draw inspiration from Lie group (Gallier (2001)) to introduce a new update paradigm. This paradigm redefines parameter updates as rotations within the original parameter space. As rotations are the rigid transformation (Daniele (2001)), the update via the $BA$ matrices solely governs the change in the rotation angle. Consequently, the unlearning process is directly driven by changes in this angle, enabling us to use the rotation angle to precisely quantify the degree of unlearning in LLMs.

Firstly, we assume a cognitive rotation space $R$. Since $R$ can be viewed as an $n$-dimensional rotation matrix, $R$ satisfies the following conditions: The $R$ is an orthogonal matrix. The determinant of $R$ is 1, so $det\left(R\right) = 1$. Since $R$ directly satisfies the two conditions of $SO(n)$, we have $R \in SO(n)$. From Gallier (2001), we can then conclude that cognitive rotation space $R \in SO(n)$ corresponds to at least one matrix $C$ in the Lie algebra $\mathfrak{so}(n)$.

However, every element $C$ in the Lie algebra $\mathfrak{so}\,(n)$ can be mapped to the $R$ in the Lie group $SO\,(n)$ by the exponential map $exp\,(C)$. From this we obtain the cognitive rotation space $R$:

$$R = exp\,(C)\,, \tag{4}$$

here, $C$ is an antisymmetric matrix. Since $R = exp\,(C)$, we can obtain from Taylor's Formula:

$$R = exp\,(C) = I + C + \frac{C^2}{2!} + ... \approx I + C, C << I. \tag{5}$$

From Equation 5, we can conclude that for any antisymmetric matrix $C$, there exists a corresponding cognitive rotation space $R_C$. Therefore, we construct the skew symmetric loss $\mathcal{L}_{Sk}$ and impose the constraint that $BA$ is an antisymmetric matrix:

$$\mathcal{L}_{Sk} = \left\| (BA)^T + BA \right\|_F^2\,, \tag{6}$$

here, $I$ is the identity matrix, and $\|\cdot\|_F^2$ is the Frobenius norm.

In addition, due to the influence of factors such as the learning rate, $BA << I$ (as summarized in Table 4 and Figure 6 of Appendix A.6, the specifications for the $BA$ matrix parameters are detailed there.). Thereby, there exists a cognitive rotation space $R_{BA} \approx I + BA$.

We can establish the following LoRA update paradigm:

$$W \leftarrow W + BAW = (I + BA)\,W\,, \tag{7}$$

here, we update the above equation given a set of low-rank parameters $\{A, B\}$, where the parameter matrix $W$ of LLM is frozen.

From Equation 7, we can consider the update of the $W$ as a rotation in the cognitive rotation space $R_{BA}$.

After expressing the update of $BA$ as the cognitive rotation space $R_{BA}$ in the parameter space, we aim to quantify the degree of unlearning of LLM by changing the rotation angle of $R_{BA}$.

Although we only trained the attention layers, due to the huge parameter size of the large language model, directly calculating the rotation angle $\theta$ of $R_{BA}$ still incurs a significant amount of computational cost. Here we present our theorem 1.

**Theorem 1:** *For $R \in \mathbb{R}^{n \times n}$, when $R = exp\,(C)$, the rotation angle $\theta$ of $R$ is directly proportional to $C$.* The proof of theorem 1 can be found in the appendix.

From **Theorem 1**, if we want to change the rotational angle $\theta$ of $R_{BA}$, we only need to make the corresponding changes to $BA$. Therefore, we obtain the rotational salience weight $\beta$ from the OOD detector and the distributional shift compensator. When $\beta \times BA$, all the rotation angles $\theta$ in $R_{BA}$ are changed to $\beta \times \theta$. Therefore, we can use the rotation Angle $\theta$ to quantify the unlearning degree of LoRA, and only need to use $BA$ for the calculation.

In addition, for achieving effective unlearning, we utilize preference optimization to update the model to accommodate random task labels or refuse-based answers such as "I don't know", which we call $y'$. For each unlearning we only train the cross-entropy loss using the unlearning dataset of our current knowledge:

$$\mathcal{L}_{CE} = -\frac{1}{N^{U,t}} \sum_{i=1}^{N^{U,t}} y_i'^{U,t} \log M_\Theta\left(x_i^{U,t}\right). \tag{8}$$

To reduce the interaction between the change of the rotation angle for each unlearning request in continuous unlearning, we make the rotation axes of each rotation perpendicular to each other. The rotation axes here refers to the subspace formed by the points that remain stationary under rotation in the high-dimensional space.

Here, we know that when unlearning request $t$, the corresponding cognitive rotation space is $R_{B_t A_t}$. Then, from request $t-1$ to request $t$, the relative rotation matrix is $\triangle R_t$. From this, we can obtain :

$$\triangle R_t = R_t \cdot R_{t-1}^T = (I + B_t A_t)\,(I + B_{t-1} A_{t-1})^T\,, \tag{9}$$

since $BA$ is an antisymmetric matrix, we have:

$$\triangle R_t = (I + B_t A_t)(I + B_{t-1} A_{t-1})^T = (I + B_t A_t)(I - B_{t-1} A_{t-1}) \tag{10}$$
$$= I + B_t A_t - B_{t-1} A_{t-1} - B_t A_t B_{t-1} A_{t-1} \approx I + B_t A_t - B_{t-1} A_{t-1},$$

We cannot directly calculate the rotation axes for the calculation because this would consume a large amount of computing resources and significantly slow down the training speed of the model. Here, we know from **Theorem 2** that when the cognitive rotation space $R_{B_{t-1} A_{t-1}}$, $R_{B_t A_t}$ and $B_{t-1} A_{t-1}$, $B_t A_t$ are mutually perpendicular, their rotation axes must be perpendicular. Therefore, we make the cognitive rotation space $R_{B_{t-1} A_{t-1}}$ of the $(t-1)$-th request and the relative rotation space $\triangle R$ relative to the $t$-th request and the $(t-1)$-th request mutually perpendicular. This ensures that the rotation angles of each unlearning request in the cognitive rotation space do not affect each other, reducing the cumulative catastrophic utility loss generated with continuous unlearning.

**Theorem 2:** when $R = exp(A)$ and $R' = exp(A')$, $A \perp A'$, then the rotation axes of $R$ and $R'$ are perpendicular to each other. The proof of theorem 2 can be found in the appendix.

We hope that $\triangle R_t = I + B_t A_t - B_{t-1} A_{t-1}$ and $R_{t-1} = I + B_{t-1} A_{t-1}$ are perpendicular to each other, then $B_t A_t - B_{t-1} A_{t-1}$ and $B_{t-1} A_{t-1}$ will also be perpendicular to each other from Equation 5 and Theorem 2. The orthogonal rotation axes loss are as follows:

$$\mathcal{L}_o = \|(W_t - W_{t-1}) \cdot W_{t-1}\|_F^2 = \|(B_t A_t - B_{t-1} A_{t-1}) \cdot (B_{t-1} A_{t-1})\|_F^2, \tag{11}$$

where $W_{t-1} = B_{t-1} A_{t-1}$ are the parameters of the lora after training on the $(t-1)$-th request. The $\|\cdot\|_F^2$ is the Frobenius norm.

In summary, the overall loss of our method is as follows:

$$\mathcal{L}_{overall} = \lambda_1 \mathcal{L}_{Sk} + \lambda_2 \mathcal{L}_o + \lambda_3 \mathcal{L}_{CE}, \tag{12}$$

here, we set $\lambda_1 = 0.1$, $\lambda_2 = 0.1$ and $\lambda_3 = 1$ on the ScienceQA dataset. We set $\lambda_1 = 0.01$, $\lambda_2 = 0.5$ and $\lambda_3 = 1$ on the TOFU dataset.

## 3.2 Unlearned Knowledge Detection

**OOD Detection.** Based on $o^3$, we turn the unlearned knowledge detection task into an OOD task by treating the unlearned dataset as In-Distribution (ID) data, and leverage a scoring mechanism to quantify the extent of unlearning.

We propose the OOD detection loss, which consists of three parts. We use the contrastive entropy loss and Masked Language Modeling (MLM) loss (Gao et al. (2025)). As shown in Figure 2, the updates of RCU exhibit an uneven characteristic, where the feature always involves continuous updates within a very small range of rotation angle changes. Given this characteristic, in order to make the output of OOD detection better align with the update pattern of RCU, we introduced the unlearning alignment loss $\mathcal{L}_{Ua}$. The $\mathcal{L}_{Ua}$ are as follows:

$$\mathcal{L}_{Ua} = \frac{1}{d^2} \left\| \frac{\hat{Z}_i^T \hat{Z}_i}{n-1} - I \right\|_F^2, \text{ where } \hat{Z}_i = \frac{Z_i}{\|Z_i\|_2}, \tag{13}$$

where $\|\cdot\|_2$ is $L_2$ norm. The $Z_i$ is the average pooled feature representation from layer i of the backbone network. The $\|\cdot\|_F^2$ denotes the Frobenius norm.

In addition, the contrastive entropy $\mathcal{L}_{CEL}$ also starts with the augmentation view generation. The $\mathcal{L}_{CEL}$ (Gao et al. (2025)) leverage random masking to generate the first view type. For a particular text instance $x$ with tokens of length $n$, we randomly select $p\%$ ($p = 15$ in our implementation) tokens and replace them with the tokens of [MASK]. The $x^*$ is the instance with the random masking. For the second contrastive view, we make use of a key encoder $F_{\Omega^{key}}$, which is initialized from the original OOD module backbone $F_\Omega$ that is a transformer consisting of $L$ attention layers : $F := f_{\omega_1} \circ \cdots f_{\omega_l} \circ \cdots f_{\omega_L}$. Then we input the original text instance $x$ and generate the second

view from $F_{\Omega^{key}}$. The $\mathcal{L}_{CEL}$ is as follow:

$$\mathcal{L}_{\text{CEL}} = -\sum_{i=1}^{N^B}\sum_{l=1}^{L}\sum_{j=1}^{N^B} \Delta(i,l,j)\log(\Delta(i,l,j)), \qquad (14)$$

$$\textbf{where } \Delta(i,l,j) = \frac{\exp\left(f_{\omega_{[1:l]}}(\boldsymbol{x}_i^*) \cdot f_{\omega_{[1:l]}^{\text{key}}}(\boldsymbol{x}_j)\right)}{\sum_{k=1}^{N^B}\exp\left(f_{\omega_{[1:l]}}(\boldsymbol{x}_i^*) \cdot f_{\omega_{[1:l]}^{\text{key}}}(\boldsymbol{x}_k)\right)},$$

here $N$ is the sample quantity of a mini-batch. And the $f_{\omega_{[1:l]}}(\boldsymbol{x}_i^*)$ is the token averaging representation of the $l$-th layer. We use MLM loss $\mathcal{L}_{MLM}$ (Jian et al. (2022)) to improve the language generation of our model:

$$\mathcal{L}_{MLM} = -\frac{1}{N^B}\sum_{i=1}^{N^B} y_i^* \log F_\Omega(x_i^*), \qquad (15)$$

where $y^*$ is the random token masking label. Here, the $\mathcal{L}_{CEL}$ focuses on the relative relationship of sample pairs. The $\mathcal{L}_{MLM}$ boosts the representation power of the generated language.

The final loss $\mathcal{L}_{OOD}$ can be:

$$\mathcal{L}_{OOD} = \mathcal{L}_{CEL} + \mathcal{L}_{MLM} + \mathcal{L}_{Ua}. \qquad (16)$$

**Distributional Shift Compensator.** We follow the method for obtaining the output of OOD detection as described in $o^3$. We utilized the Mahalanobis distance and the distance based on the maximum instance cosine similarity. Finally, we calculated the combined score $\gamma^t$. For the calculation of the OOD score $\gamma^t$, please refer to the appendix.

After we get combined score $\gamma^t$, we need to map the $\gamma^t$ into an rotational salience weight $\beta$. Here, we hope that the change of $\beta$ can conform to the unlearning process of the cognitive rotation space $R_{BA}$. However, as the unlearning learning proceeds, we find that the performance of the update based on RCU is uneven. As shown in the Figure 2 (a)(b), the model does not learn unlearning knowledge before $\beta = 0.3$ on the ScienceQA dataset. At $\beta = 0.3$ to $\beta = 0.5$, the unlearned knowledge is gradually learned, while at $\beta = 0.5$ to $\beta = 1$, the model has fully learned the unlearned knowledge. On the TOFU dataset, the range of knowledge that the model learns for unlearning is approximately between $\beta = 0.2$ and $\beta = 0.6$. The specific results can be found in the appendix. This gives us the following relation:

$$\beta = \begin{cases} 0.45 & \Gamma_2 < \gamma^t \leq 1, \\ \mathcal{M}(\gamma^t) & \Gamma_1 < \gamma^t \leq \Gamma_2, \\ 0 & \gamma^t \leq \Gamma_1, \end{cases} \qquad (17)$$

the $\Gamma_1$ and $\Gamma_3$ are thresholds, which $\Gamma_1 = 1e-80$, $\Gamma_2 = 0.1$ on the ScienceQA dataset and $\Gamma_1 = 0.2$, $\Gamma_2 = 1$ on the TOFU dataset. The $\mathcal{M}(\gamma^t)$ on the ScienceQA dataset is $0.35 + (\log_{10}\gamma^t + 80)/790$. The $\mathcal{M}(\gamma^t)$ on the TOFU dataset is $0.35 + ((\gamma^t - 0.2)/0.8) \cdot 0.25$.

Finally, for each input $x$ of the $t$-th unlearning request, the corresponding parameters $W_x^t$ can be expressed as:

$$W_x^t = (I + \beta \cdot BA)W. \qquad (18)$$

## 4 EXPERIMENTS

### 4.1 DATASETS

We conducted experiments on two tasks: question answering and fictional knowledge generation. We have divided the question-answering task into 5 consecutive sub-tasks, and the fictional knowledge generation has been divided into 3 consecutive sub-tasks. The detailed introduction of the datasets is as follows:

**Question Answeing.** We use ScienceQA (Lu et al. (2022)) as the question and answer dataset. This dataset consists of 6,508 training samples and 2,224 testing samples. We selected five of

these areas as the continuous unlearning requests, namely biology, physics, chemistry, economics, and earth-science. We utilized the CommonsenseQA (Talmor et al. (2018)) as the utility dataset, which contained 9,740 training samples and 1,221 validation samples, to evaluate the commonsense reasoning ability of LLMs. The OpenbookQA (Taori et al. (2023)) can assess the understanding ability of books. The training set contains 4,957 samples, the validation set includes 500 samples, and the test set consists of 500 samples.

**Fictitious Knowledge Generation.** We conducted a test on the generation of fictional knowledge using the TOFU dataset (Maini et al. (2024)). The TOFU dataset contains questions about fictional authors synthesized by GPT-4. The three unlearning sets 'foget01', 'foget05', and 'foget10' respectively represent random selection ratios of 1%, 5%, and 10% of the authors. The authors in each unlearning set are mutually exclusive. Additionally, we also utilized the data related to real-word authors and world facts in this dataset to test the LLMs' ability to maintain its effectiveness.

## 4.2 EXPERIMENTAL SETUP

Table 1: Performance Comparison between our method and other baselines when continually unlearning TOFU-forget01, -forget05, and -forget10 in Fictitious Knowledge Generation. The * represents the results we achieved in our own experimental environment.

| Method | Unlearning Request 1 | | | | | Unlearning Request 2 | | | | | Unlearning Request 3 | | | | |
|---|---|---|---|---|---|---|---|---|---|---|---|---|---|---|---|
| | S.U.↓ | D.U.↓ | R.D.↑ | R.A.↑ | W.F.↑ | S.U.↓ | D.U.↓ | R.D.↑ | R.A.↑ | W.F.↑ | S.U.↓ | D.U.↓ | R.D.↑ | R.A.↑ | W.F.↑ |
| Base * | 85.0 | 90.0 | 85.8 | 89.0 | 87.0 | 87.3 | 89.3 | 85.8 | 89.0 | 87.0 | 85.3 | 90.0 | 85.8 | 89.0 | 87.0 |
| GradASC | 75.0 | 85.0 | 71.0 | 86.0 | 82.1 | 17.6 | 23.1 | 19.0 | 0 | 0 | 17.1 | **14.2** | 19.0 | 0 | 0 |
| GradDif | 78.1 | 84.0 | 81.9 | 86.7 | 83.5 | 62.5 | 70.0 | 70.4 | 65.7 | 77.9 | **16.5** | 15.2 | 19.0 | 0 | 0 |
| EUL | 84.1 | 86.3 | **86.1** | 86.7 | 87.1 | 84.4 | 90.3 | **85.8** | 88.0 | 85.5 | 80.1 | 83.5 | 83.4 | 86.3 | 83.5 |
| PO * | 18.75 | 25.0 | 77.0 | 85.0 | 81.0 | 31.88 | 52.5 | 79.0 | 86.0 | 79.0 | 43.13 | 52.5 | 77.75 | 78.0 | 80.0 |
| NPO | 68.8 | 75.0 | 83.6 | 89.0 | 81.8 | 76.3 | 84.2 | 83.2 | 87.7 | 84.1 | 77.6 | 79.2 | 81.4 | 87.3 | 82.9 |
| SOGD | 43.7 | 76.0 | 80.3 | 85.3 | 83.4 | 22.8 | 24.0 | 79.0 | 81.3 | 82.6 | 17.4 | 21.7 | 82.3 | 77.0 | 82.1 |
| SOPO * | 31.25 | 37.5 | 83.7 | 85.0 | 83.0 | 38.13 | 45.0 | 80.0 | 87.0 | 82.0 | 36.88 | 43.75 | 79.5 | 85.0 | 82.0 |
| $o^3$ * | 15.66 | 14.67 | 85.25 | 89.0 | 86.3 | 22.49 | 20.17 | 85.5 | 89.0 | 86.3 | 26.66 | 23.56 | 85.25 | 89.0 | 86.3 |
| Ours | 9.37 | 12.5 | 85.57 | **89.0** | **87.0** | 12.5 | 12.5 | 85.61 | **89.0** | **87.0** | 20.6 | 17.5 | **85.6** | **89.0** | **87.0** |

**Evaluation Metrics.** We continue to use the $o^3$ (Gao et al. (2025)) key evaluation indicators. Here, the Sample-level Unlearning (S.U.) represents the performance of the test LLMs on the unlearning training set when there is a unlearning request. The Distribution-level Unlearning (D.U.) indicates the performance of the test LLMs on the unlearning test set when there is the unlearning request. In addition, we use three indicators to measure the performance of the LLMs in maintaining utility. The Retained Distribution (R.D.) represents the distribution that is most sensitive to unlearning requests. The CommonsenseQA (C.QA.) and QpenbookQA (O.QA.) are datasets used on the scienceQA dataset to measure the utility of each request. We use the accuracy of these two datasets to measure their performance in QA. On the TOFU dataset, we use the utility datasets provided by the two datasets, namely Real-word Authors (R.A.) and Word Fact (W.F.), to measure their performance in Fictitious Knowledge Generation.

**Compared Baseline.** We compared a series of the most advanced LLM unlearning methods: GradAsc Golatkar et al. (2020), GradDif Yao et al. (2024), EUL Chen & Yang (2023a), PO Eldan & Russinovich (2023), NPO Zhang et al. (2024), SOGD Jia et al. (2024), SOPO Jia et al. (2024) and $o^3$ Gao et al. (2025). The base refers to the result obtained directly through the LLM testing.

**Implementation Details.** We use LLaMA2-7b (Touvron et al. (2023)) as the target model. The detection backbone is Pseudo-Roberta-Large (Liu et al. (2019)). For the TOFU dataset, the learning rate is 2e-4, and the number of epoch is 10. For the ScienceQA dataset, the batch size is 128, the number of epoch is 15, and the learning rate is 3e-4. In the inference pipeline of ScienceQA, the max batch size is 24. The LoRA ranks for both datasets are 8. Our method only fine-tunes the attention layers in LLM. We conducted our experiments on two NVIDIA RTX A6000.

## 4.3 EXPERIMENTAL RESULTS

**Question Answering.** The results of our method are shown in appendix A.8 and Figure 3. We compared base, PO, SOPO, $o^3$ and our results. Some detailed results can be found in the appendix. Our method achieved the same results as the base method (C.QA. and O.QA.). The results of the

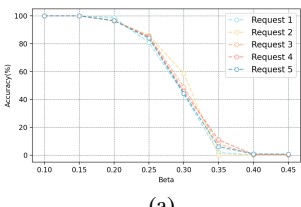 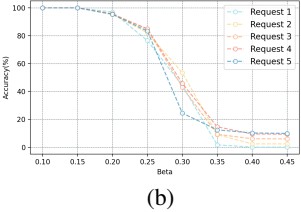 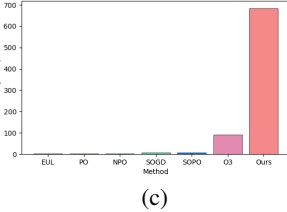

(a)                                    (b)                                    (c)

Figure 2: Experimental results on the ScienceQA dataset. (a) The relationship between the $\beta$ ($\beta$ is proportional to the rotation angle) and unlearning processes (S.U.). (b) The relationship between the $\beta$ ($\beta$ is proportional to the rotation angle) and unlearning processes (D.U.). (c) The results of $U^2R$ on the TOFU dataset.

R.D. were also very similar to those of the base method. However, our method performed much better than the others in the S.U. and D.U. indicators. Compared with the most advanced method $o^3$, the S.U. decreased by 9.68%, 4.02%, 4.05%, 7.58% and 10.82% respectively on five requests. The D.U. compared to the most advanced method $o^3$, on average, decreased by 17.67% across five requests. In addition, the number of training parameters of our method is much smaller than that of $o^3$.

**Fictitious Knowledge Generation.** Table 1 presents the experimental results of our method on the TOFU dataset. Based on the results, we found that our method can effectively enhance the LLMs' ability to unlearning (D.U. and S.U.). Compared with the currently best continuous unlearning method $o^3$, the result on the S.U. decreased by 6.29%, 9.99% and 6.06%. the result on the D.U. decreased by 2.17%, 7.67% and 6.06%. Furthermore, in terms of the stability of the LLMs' utility, the performance level of our method is comparable to the current best results. We observed that as the number of unlearning requests increased, the unlearning accuracy on the test set (D.U.) of most methods showed significant fluctuations or increasing. This phenomenon indicates that catastrophic utility loss continues to accumulate, and the LLMs' stability is affected. In contrast, our method not only effectively mitigates such utility losses but also significantly improves the performance stability of LLMs during the continuous unlearning process.

In two datasets, $o^3$ has 19.99 M trainable parameters, compared to our method's 8.39 M. The required number of training parameters is significantly less than that of $o^3$. The number of parameters that need to be updated in our method is much smaller than that in $o^3$.

Furthermore, we follow the Unlearning-Utility Ratio ($U^2R$) in $o^3$.

$$U^2R = \frac{\text{Acc}^0_{\text{S.U.}} + \text{Acc}^0_{\text{D.U.}} - \text{Acc}^T_{\text{S.U.}} - \text{Acc}^T_{\text{D.U.}}}{\text{Acc}^0_{\text{R.D.}} + \text{Acc}^0_{\text{U.1.}} + \text{Acc}^0_{\text{U.2.}} - \text{Acc}^T_{\text{R.D.}} - \text{Acc}^T_{\text{U.1.}} - \text{Acc}^T_{\text{U.2.}}}, \tag{19}$$

where the Acc denotes Accuracy, the U.1 represents C.QA or R.A., the U.2 represents O.QA or W.F., and the T denotes the unlearning request. The result on the TOFU dataset is shown in Figure 2 (c). Our method has demonstrated significant advantages in both unlearning and utility retention. Furthermore, the results on the ScienceQA dataset can be found in the Appendix A.2.

## 4.4 THE RELATIONSHIP BETWEEN ROTATION AND UNLEARNING

In Figure 2 (a) and Figure 2 (b), we have presented the changes about $\beta$, along with the experimental results for each unlearning request. Based on the results on the ScienceQA dataset, we can observe that when $\beta$ is within the range of 0.15 to 0.45, the model rapidly undergoes the process of unlearning. However, when $\beta$ is less than 0.15, the model does not forget the knowledge. And when $\beta$ is greater than 0.45, the model has achieved complete forgetting and no longer continues to unlearning. The above experimental results indicate that the unlearning process of RCU is achieved within a very small rotation angle change range in the cognitive rotation space. Therefore, when designing the OOD detector and the distributional shift compensator, we must also ensure that the

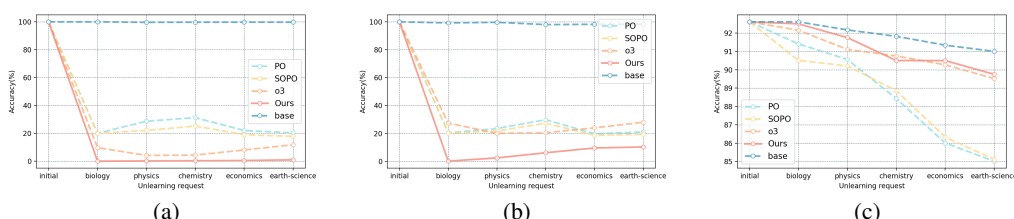

(a)             (b)             (c)

Figure 3: The comparison results with other methods on the ScienceQA dataset. (a) The results of S.U.. (b) The results of D.U.. (c) The results of R.D..

rotational salience weights they output have a relatively concentrated distribution to match the above characteristics. The $\beta$ changes regarding TOFU dataset are presented in the appendix.

## 4.5 ABLATION STUDY

The results of our ablation experiments are shown in Table 2. Here, we aim to focus more on the unlearning process rather than the OOD process. Therefore, we consider the impact of $\mathcal{L}_{Ua}$ on the unlearning performance rather than on the OOD performance. Based on the results, we found that $\mathcal{L}_{Sk}$ is the main influencing factor for the unlearning performance. And the LoRA update Paradigm (RC-LoRA in Table 2), $\mathcal{L}_{Sk}$ and the $\mathcal{L}_o$ are three key factors for mitigating cumulative catastrophic utility loss in continuous unlearning. The results of the ablation experiment proved the effectiveness of our method.

Table 2: The ablation experiment results on the ScienceQA dataset.The RC-LoRA represents our LoRA update paradigm.

| Method | Unlearning Request 1 | | | | | Unlearning Request 2 | | | | | Unlearning Request 3 | | | | |
|---|---|---|---|---|---|---|---|---|---|---|---|---|---|---|---|
| | S.U.↓ | D.U.↓ | R.D.↑ | C.QA.↑ | O.QA.↑ | S.U.↓ | D.U.↓ | R.D.↑ | C.QA.↑ | O.QA.↑ | S.U.↓ | D.U.↓ | R.D.↑ | C.QA.↑ | O.QA.↑ |
| w/o $\mathcal{L}_o$ | 0.54 | 0.68 | 89.01 | 79.19 | 83.09 | 0.65 | 4.74 | 88.02 | 79.19 | 83.09 | 0.78 | 8.98 | 88.22 | 79.19 | 83.09 |
| w/o $\mathcal{L}_{Sk}$ | 99.75 | 98.99 | 90.16 | 79.19 | 83.09 | 99.51 | 98.01 | 89.18 | 79.19 | 83.09 | 99.51 | 98.04 | 88.44 | 79.19 | 83.09 |
| w/o RC-LoRA | 0.12 | 1.35 | 90.51 | 79.19 | 82.39 | 0.58 | 4.24 | 91.11 | 79.19 | 82.39 | 0.99 | 22.45 | 90.68 | 79.19 | 82.39 |
| w/o $\mathcal{L}_{Sk}$+RC-LoRA | 100.00 | 99.24 | 89.16 | 79.19 | 83.09 | 99.52 | 98.14 | 88.25 | 79.19 | 83.09 | 99.62 | 98.53 | 87.44 | 79.19 | 83.09 |
| w/o $\mathcal{L}_{Ua}$ | 1.27 | 3.44 | 87.01 | 79.02 | 81.35 | 2.77 | 4.85 | 88.08 | 79.02 | 81.35 | 3.11 | 8.23 | 87.62 | 79.02 | 81.35 |
| **Ours** | **0.05** | **0.12** | **92.5** | **79.19** | **83.2** | **0.17** | **2.45** | **91.75** | **79.19** | **83.2** | **0.33** | **6.13** | **90.5** | **79.19** | **83.2** |

| Method | Unlearning Request 4 | | | | | Unlearning Request 5 | | | | |
|---|---|---|---|---|---|---|---|---|---|---|
| | S.U.↓ | D.U.↓ | R.D.↑ | C.QA.↑ | O.QA.↑ | S.U.↓ | D.U.↓ | R.D.↑ | C.QA.↑ | O.QA.↑ |
| w/o $\mathcal{L}_o$ | 1.24 | 11.58 | 88.06 | 79.19 | 83.09 | 2.41 | 16.85 | 87.95 | 79.19 | 83.09 |
| w/o $\mathcal{L}_{Sk}$ | 99.75 | 98.04 | 88.20 | 79.19 | 83.09 | 99.75 | 98.04 | 87.95 | 79.19 | 83.09 |
| w/o RC-LoRA | 1.03 | 30.68 | 89.43 | 79.19 | 83.09 | 2.46 | 31.85 | **90.00** | 79.19 | 83.09 |
| w/o $\mathcal{L}_{Sk}$+RC-LoRA | 99.85 | 98.53 | 87.2 | 79.19 | 83.09 | 99.85 | 98.28 | 86.45 | 79.19 | 83.09 |
| w/o $\mathcal{L}_{Ua}$ | 9.19 | 11.07 | 87.33 | 79.02 | 81.35 | 13.65 | 15.21 | 86.93 | 79.02 | 81.35 |
| **Ours** | **0.49** | **9.56** | **90.5** | **79.19** | **83.2** | **0.98** | **10.29** | 89.753 | **79.19** | **83.2** |

## 5 CONCLUSION

Existing LLM unlearning methods are vulnerable not only to cumulative catastrophic utility loss from continuous unlearning requests, but also to a fundamental limitation in practicality owing to their heavy reliance on the retained dataset. To overcome this limitation, we introduce RCU, a retain-free approach that formulates LoRA updates as rotations within a specially constructed cognitive rotation space. This formulation makes the rotational angle updated by LoRA the sole variable, which is strongly correlated with the degree of unlearning. Consequently, the unlearning process can be directly quantified by corresponding changes in the rotational angle. Furthermore, by analyzing the update dynamics during unlearning, we propose a rotational salience weight to achieve precise and continuous control over the unlearning process. Our method is supported by theoretical guarantees, and we rigorously establish its efficacy through a mathematical analysis. Extensive experiments demonstrate that RCU achieves superior unlearning effectiveness while maintaining model utility on multiple datasets.

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

## A APPENDIX

This Appendix includes additional details for the our paper including the following aspects:

- A.1: The proofs of the theorem 1 and theorem 2.

- A.2: Detailed experimental results on the ScienceQA dataset.

- A.3: Large Language Model usage declaration.

- A.4: The experimental results on the TOFU dataset showing the relationship between $\beta$ and the unlearning process.

- A.5: The detailed calculation steps of the OOD Score.

- A.6: An in-depth study on why our method is effective.

- A.7: Experimental results regarding the hyperparameter $\lambda$

- A.8: Computation overhead analysis.

- A.9: The comparison results of the Truth Ratio on the TOFU dataset.

## A.1 Proofs of the Theorems.

**Theorem 1:** *For $R \in \mathbb{R}^{n \times n}$, when $R = exp\,(C)$, the rotation angle $\theta$ of $R$ is directly proportional to $C$.*

**Proof 1:** It is known that $R \in SO\,(n)$ is an n-dimensional rotation matrix ($n > 3$) and $R = exp\,(C)$, where $C$ is an antisymmetric matrix. If the rotation angle of $R = exp\,(C)$ is $\theta$, then the rotation Angle of $exp\,(kC)$ is $k\theta$.

Due to $R \in SO\,(n)$, there exist orthogonal matrices $Q$ such that:

$$R = Q \cdot diag\,(1, ..., 1.R\,(\theta_1), ..., R\,(\theta_m)) \cdot Q^T, \tag{20}$$

here, $R = \begin{bmatrix} \cos \theta_j & -\sin \theta_j \\ \sin \theta_j & \cos \theta_j \end{bmatrix}$ is Two-dimensional rotation matrix (the rotation angle is $\theta_j$ ), the rest of the eigenvalues are 1. The $m = [n/2]$.

The antisymmetric matrix $C$ can be similarly block-diagonalized as follows:

$$C = Q \cdot diag\,(0, ..., 0, B\,(\theta_1), ..., B\,(\theta_m)) \cdot Q^T, \tag{21}$$

where $B = \begin{bmatrix} 0 & -\theta_j \\ \theta_j & 0 \end{bmatrix}$ . $exp\,(B\,(\theta_j)) = R\,(\theta_j)$.

If $C' = kC$ ,we can gather:

$$C' = Q \cdot diag\,(0, ..., 0, B\,(\theta_1), ..., B\,(\theta_m)) \cdot Q^T, \tag{22}$$

since B is linear, $B\,(k\theta_j) = kB\,(\theta_j)$. Then:

$$exp\,(kC) = exp\,(C') = Q \cdot exp\,(diag\,(0, ..., 0, B\,(k\theta_1), ..., B\,(k\theta_m))) \cdot Q^T \tag{23}$$

$$= Q \cdot diag\,(1, ..., 1.B\,(k\theta_1), ..., B\,(k\theta_m)) \cdot Q^T.$$

And the $exp\,(B\,(k\theta_j))$ is as follow:

$$exp\,(B\,(k\theta_j)) = \begin{bmatrix} \cos\,(k\theta_j) & -\sin\,(k\theta_j) \\ \sin\,(k\theta_j) & \cos\,(k\theta_j) \end{bmatrix} = R\,(k\theta_j). \tag{24}$$

Due to the Equation 25 and Equation 24, we can get as follow:

$$exp\,(kC) = Q \cdot diag\,(0, ..., 0, R\,(k\theta_1), ..., R\,(k\theta_m)) \cdot Q^T, \tag{25}$$

here means the rotation angle of $exp\,(kC)$ are $k\theta_1, ..., k\theta_m$.

**Theorem 2:** when $R = exp\,(A)$ and $R' = exp\,(A')$ , $A \perp A'$, then the rotation axes of $R$ and $R'$ are perpendicular to each other.

**Proof 2:** Let $A$ and $A'$ be skew-symmetric matrices with their rotation faces $P$ and $P'$, respectively. Assume that $P$ and $P'$ are orthogonal to each other ($A$ and $A'$ are perpendicular). Since $P \perp P'$, there are $P \subseteq ker\,(A')$ and $P' \subseteq ker\,(A)$. (Here, $ker\,(\cdot)$ refers to the null space.) Thus, for any vector $v$, $A'Av = 0$ and $AA'v = 0$ , which is $AA' = A'A = 0$.

Now, $R = exp\,(A) = I + A + \frac{A^2}{2!} + ....$ Similarly, $R' = exp\,(A')$.

The rotation face of $R$ is $P$ (the eigenspace corresponding to the nonzero eigenvalues of $A$), since $exp\,(A)$ is the usual rotation on $P$ and identity elsewhere. Similarly, the surface of rotation of $R'$ is $P'$. Since $P$ and $P'$ are orthogonal, the spaces of rotation of $R$ and $R'$ are perpendicular to each other.

Thus, when the rotation faces of the skewsymmetric matrices $A$ and $A'$ generating rotations are perpendicular to each other, the rotation spaces of the corresponding rotation matrices $R'$ and $R$ are also perpendicular to each other.

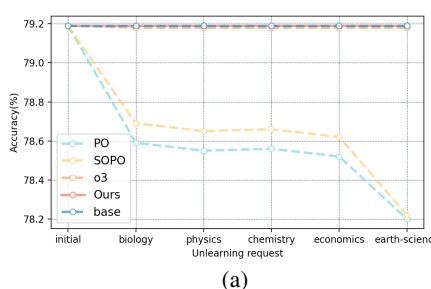 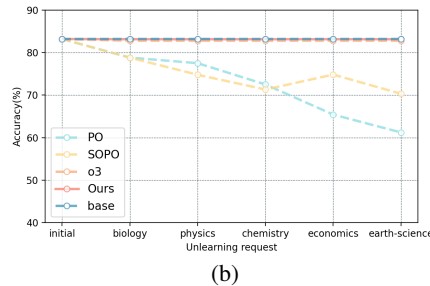

| (a) | (b) |

Figure 4: Experimental results on the ScienceQA dataset. (a) The results of C.QA.. (b)The results of O.QA..

## A.2  MORE EXPERIMENTAL RESULTS

The detailed results of our work on the ScienceQA dataset are shown in Table 3. Based on the results, we can observe that compared to $o^3$, our method is more stable and has a better ability to resist cumulative catastrophic utility loss. Here, the $U^2R$ of our method is 65.79, and the $U^2R$ of the other method is 48.46.

Table 3: Performance Comparison between our method and other baselines when continually unlearning biology, physics, chemistry, economics and earth-science in Fictitious Knowledge Generation. The unlearning effectiveness is measured by the generation accuracy of the unlearning train data and unlearning test data denoted as S.U. and D.U., CommonsenseQA (C.QA.), OpenbookQA (O.QA.) respectively. Utility preservation is evaluated by the generation accuracy of Retained Distribution (R.D.). The * represents the results we achieved in our own experimental environment.

| Method | Unlearning Request 1 | | | | | Unlearning Request 2 | | | | | Unlearning Request 3 | | | | |
|---|---|---|---|---|---|---|---|---|---|---|---|---|---|---|---|
| | S.U.↓ | D.U.↓ | R.D.↑ | C.QA.↑ | O.QA.↑ | S.U.↓ | D.U.↓ | R.D.↑ | C.QA.↑ | O.QA.↑ | S.U.↓ | D.U.↓ | R.D.↑ | C.QA.↑ | O.QA.↑ |
| Base * | 100 | 99.24 | 92.61 | 79.19 | 83.2 | 99.66 | 98.3 | 92.17 | 79.192 | 83.2 | 99.73 | 98.02 | 91.82 | 79.19 | 83.2 |
| $o^3$ * | 9.73 | 27.4 | 92.15 | 79.19 | 83.0 | 4.19 | 20.03 | 91.11 | 79.19 | 83.0 | 4.38 | 20.48 | 90.25 | 79.19 | 83.0 |
| Ours | **0.05** | **0.12** | **92.5** | **79.19** | **83.2** | **0.17** | **2.45** | **91.75** | **79.19** | **83.2** | **0.33** | **6.13** | **90.5** | **79.19** | **83.2** |

| Method | Unlearning Request 4 | | | | | Unlearning Request 5 | | | | |
|---|---|---|---|---|---|---|---|---|---|---|
| | S.U.↓ | D.U.↓ | R.D.↑ | C.QA.↑ | O.QA.↑ | S.U.↓ | D.U.↓ | R.D.↑ | C.QA.↑ | O.QA.↑ |
| Base | 99.75 | 98.23 | 91.34 | 79.19 | 83.2 | 99.77 | 98.25 | 91.0 | 79.19 | 83.2 |
| $o^3$ | 8.07 | 23.98 | 90.17 | 79.19 | 83.0 | 11.8 | 28.0 | 89.52 | 79.19 | 83.0 |
| Ours | **0.49** | **9.56** | **90.5** | **79.19** | **83.2** | **0.98** | **10.29** | **89.753** | **79.19** | **83.2** |

The specific results are shown in Figure 4.

## A.3  LARGE LANGUAGE MODEL USAGE DECLARATION

During this research process, the LLM provided significant assistance in organizing the logic of the paper and improving the language expression. Here, we express our gratitude for the role that the LLM played in enhancing the logicality and clarity of this research.

## A.4  HYPERPARAMETER ANALYSIS ON THE TOFU DATASET

The results on TOFU dataset are shown in Figure 5. We found that the change of $\beta$ shifted slightly to the right for a short period, but the overall change was still concentrated in a certain area. Therefore, when designing the distributional shift compensator for TOFU dataset, we also tried to map $\mathcal{M}(\gamma^t)$ to the range of 0.2 to 0.6.

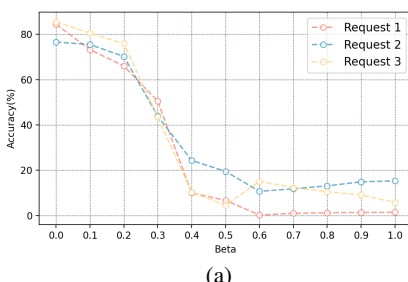 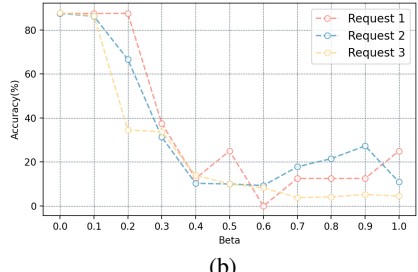

(a)  (b)

Figure 5: The relationship between the $\beta$ process and the unlearning process. (a) The results of S.U. on the TOFU dataset. (b) The results of D.U. on the TOFU dataset.

### A.5 CALCULATION OF OOD SCORE

We follow the method for obtaining the output of OOD detection as described in $o^3$. Our method for calculating the OOD score is as follows:

$$s(x)_l = \left(f_{w[1:l]} - \mu_l\right)^T \sum_l^{-1} \left(f_{w[1:l]} - \mu_l\right) + \gamma \cdot \left(-\max_{i=1}^{\alpha N^U} \left\{\frac{f_{w_{[1:l]}}(x) \cdot f_{w_{[1:l]}}\left(x_i^U\right)}{\left|f_{w_{[1:l]}}(x)\right|\left|f_{w_{[1:l]}}\left(x_i^U\right)\right|}\right\}\right), \tag{26}$$

$$\mu_l = \frac{1}{\alpha N^U}\sum_{i=1}^{\alpha N^U} f_{\omega_{[1:l]}}\left(\boldsymbol{x}_i^U\right), \Sigma_l = \frac{1}{\alpha N^U}\sum_{i=1}^{\alpha N^U}\left(f_{\omega_{[1:l]}}\left(\boldsymbol{x}_i^U\right) - \mu_l\right)\left(f_{\omega_{[1:l]}}\left(\boldsymbol{x}_i^U\right) - \mu_l\right)^\top, \tag{27}$$

here, $r = 1000$, $f_{w_l}$ representing the parameter of layer $l$. $D_{used}^{U,t}$ refers to one of the two subsets randomly divided from the training dataset $D^{U,t}$, which contains $\alpha N^{U,t}$ samples.

When the $T$-th re-learning request is completed, each test input $x$ is input into the OOD detection to calculate the score vector, and the distance between $x$ and the hyper-spherical $\mathcal{H}^t\left(c^t, r^t\right)$ boundary is obtained using one-class SVM. The final score is:

$$d_{\mathcal{H}^\sqcup}(x) = \left|s(x)^t - c^t\right| - r^t, \tag{28}$$

$$\gamma^t = \delta\left\{\zeta\left[1 - \max(p, p') + \min(p, p')\right]\right\}, p = \mathcal{P}_{\text{mix}}^t\left(\,d_{\mathcal{H}^t}(\boldsymbol{x})\right), p' = \mathcal{P}_{\text{mix}}^t\left(2\,d_{\mathcal{H}^t}^0 - d_{\mathcal{H}^t}(\boldsymbol{x})\right), \tag{29}$$

here, $c^t$ and $r^t$ represent the center vector and radius of the hypersphere. $\mathcal{P}_{\text{mix}}$ is the mixed gaussian distribution function.

### A.6 RESEARCH ON OUR METHOD

Starting from the intrinsic characteristics of the model parameters, we analyzed the proposed method. Specifically, the magnitude of the learnable parameter $\Theta_{BA}$ in each training round is shown in Table 4. The study found that when the parameters $\Theta_{BA} \ll 1$, the matrix $I + BA$ always has a corresponding cognitive rotation space. In the ablation experiments (Table 2), when we remove the designed LoRA update paradigm, the unlearning effect measured by D.U. is relatively ideal when

Table 4: On the ScienceQA dataset, the parameter $\Theta_{BA}$ maintained a stable performance at the $10^{-6}$ scale in multiple unlearning requests (block 1 in attention layers)

| Unlearning Request | Param $B$ | Param $A$ |
|---|---|---|
| biology | $10^{-6}$ | $10^{-5}$ |
| physics | $10^{-5}$ | $10^{-5}$ |
| chemistry | $10^{-5}$ | $10^{-5}$ |
| economics | $10^{-6}$ | $10^{-5}$ |
| earth-science | $10^{-6}$ | $10^{-5}$ |

the unlearning request is 1 to 2. How-
ever, when the unlearning request reached 3 or higher, due to the cumulative catastrophic utility
loss, the unlearning performance reflected by D.U. significantly decreased. The core of this method
lies in introducing the Lsk loss, which can constrain the parameters $\Theta_{BA}$ to continuously maintain a
small amplitude, thereby maintaining the effectiveness of the cognitive rotation space and ensuring
the continuous efficacy of the method in handling the continuous unlearning. The results shown in
Table 4 pertain to the queries layer of the attention layers (block 1). The other results generally vary
within the range of $10^{-6}$ to $10^{-5}$.

Furthermore, Figure 6 (a)(b) present the evolution of parameters $\Theta_A$ and $\Theta_B$ throughout the experi-
ment. The results indicate that the magnitudes of both $\Theta_A$ and $\Theta_B$ remain significantly smaller than
those of parameter I during the entire training process.

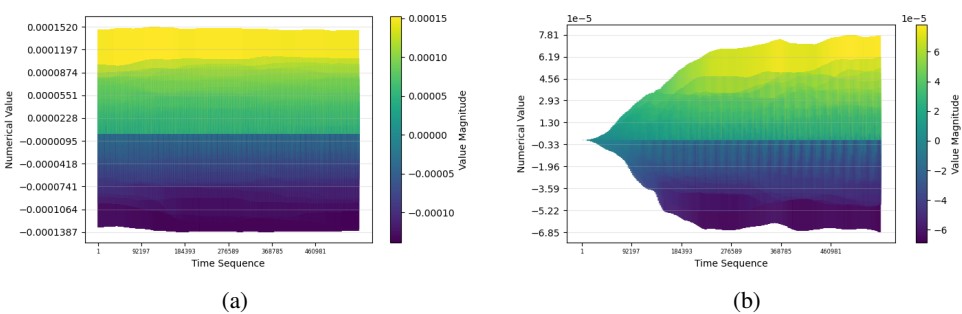

(a)                                                    (b)

Figure 6: The results of 5 continuous unlearning processes on the ScienceQA dataset. (a) The
changes in the $\Theta_A$. (b) The changes in the $\Theta_B$.

## A.7 HYPERPARAMETER ANALYSIS

We conducted a comprehensive hyperparameter analysis on $\lambda_1$, $\lambda_2$, and $\lambda_3$ across two datasets. The
results, presented in Figure 7 and Figure 8, demonstrate that our chosen hyperparameter configura-
tion consistently achieves optimal performance.

## A.8 COMPUTATION OVERHEAD ANALYSIS.

During the reasoning process, the computational cost for LLM to update using the lora update
paradigm proposed by us is 45.10 GFLOPs, the computational cost for OOD Detection is 709
MFLOPs, and the computational cost for the Distributional Shift Compensator is 13255 MFLOPs.
The total computational cost is 59.064 GFLOPs. In addition, our method requires less storage space.
The storage space required by lora in $o^3$ is 39MB, while our method only requires 16MB, reducing
the additional storage space requirement by 58%.

## A.9 THE COMPARISON RESULTS OF THE EVALUATION INDICATORS ON THE TOFU DATASET.

We choose the Truth Ratio and ROUGE-L as the indicators for evaluating the utility on the TOFU
dataset.

The results of the Truth Ratio (Maini et al. (2024)) are presented in Table 5. From these results, it
can be seen that our method is superior to the comparison methods. The formula for the Truth Ratio
is as shown in Equation 30.

$$R_{\text{truth}} = \frac{\frac{1}{\mathcal{A}_{\text{pert}}} \sum_{\hat{a} \in \mathcal{A}_{\text{pert}}} P(\hat{a} \mid q)^{1/|\hat{a}|}}{P(\tilde{a} \mid q)^{1/|\tilde{a}|}}, \tag{30}$$

where $\tilde{a}$ is a paraphrased version of the correct original answer $a$, $\tilde{a} \in \mathcal{A}_{pert}$ are deliberately per-
turbed (incorrect) answers derived from $\tilde{a}$, and $|\tilde{a}|$ denotes the number of tokens in $\tilde{a}$.

We compute the ROUGE-L recall score (Maini et al. (2024)), which acts as a surrogate for accuracy
on the question answering task, as it accounts for the output phrasing to be slightly different than

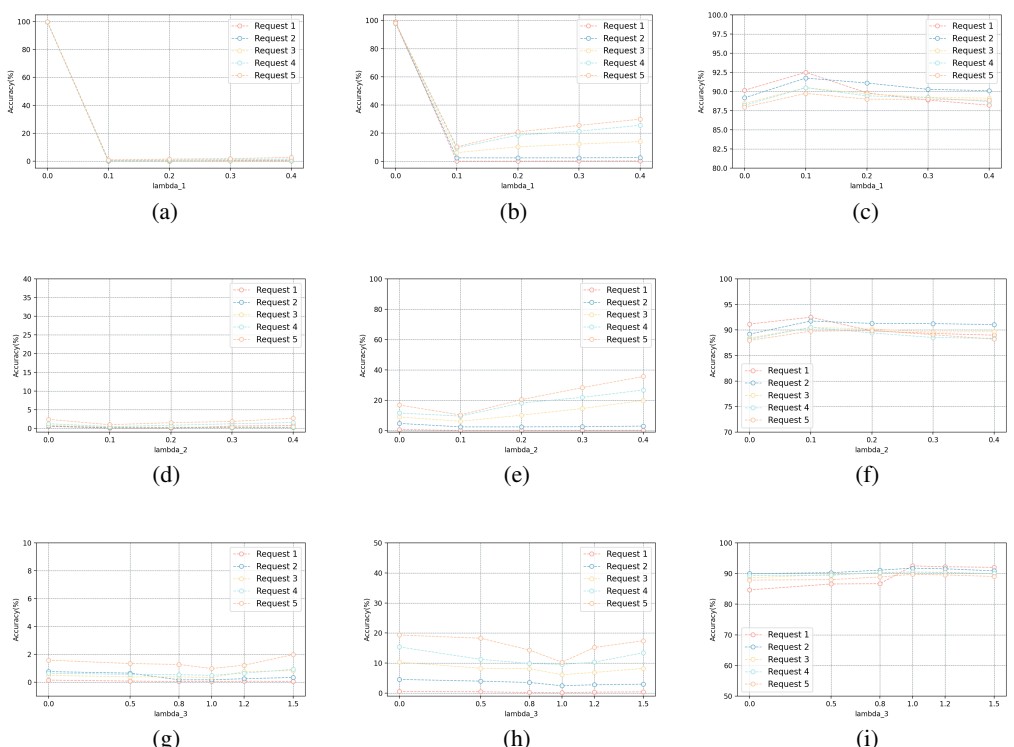

Figure 7: Experimental results on the ScienceQA dataset. The left column shows the results of S.U., the middle column shows the results of D.U., and the right column shows the results of R.D.. (a) (b) (c)The result of the hyperparameter $\lambda_1$. (d) (e) (f)The result of the hyperparameter $\lambda_2$(S.U.). (g) (h) (i)The result of the hyperparameter $\lambda_3$(S.U.).

Table 5: The results here are all the Truth Ratio (Maini et al. (2024)) corresponding to the aforementioned indicators.

| Method | Unlearning Request 1 | | | | | Unlearning Request 2 | | | | | Unlearning Request 3 | | | | |
|---|---|---|---|---|---|---|---|---|---|---|---|---|---|---|---|
| Truth Ratio | S.U.↑ | D.U.↑ | R.D.↓ | R.A.↓ | W.F.↓ | S.U.↑ | D.U.↑ | R.D.↓ | R.A.↓ | W.F.↓ | S.U.↑ | D.U.↑ | R.D.↓ | R.A.↓ | W.F.↓ |
| $o^3$ | 0.74 | 0.66 | 0.57 | 0.54 | 1.54 | 0.65 | 0.65 | 0.57 | 0.54 | 1.54 | 0.66 | 0.65 | 0.57 | 0.54 | 1.54 |
| Ours | **1.00** | **1.11** | **0.56** | **0.54** | **1.22** | **0.99** | **0.97** | **0.56** | **0.54** | **1.22** | **0.97** | **1.01** | **0.56** | **0.54** | **1.22** |

the ground truth. The results of the ROUGE-L (Maini et al. (2024)) are presented in Table 6. From these results, it can be seen that our method is superior to the comparison methods.

Table 6: The results here are all the ROUGE-L (Maini et al. (2024)) corresponding to the aforementioned indicators.

| Method | Unlearning Request 1 | | | | | Unlearning Request 2 | | | | | Unlearning Request 3 | | | | |
|---|---|---|---|---|---|---|---|---|---|---|---|---|---|---|---|
| ROUGE-L | S.U.↓ | D.U.↓ | R.D.↑ | R.A.↑ | W.F.↑ | S.U.↓ | D.U.↓ | R.D.↑ | R.A.↑ | W.F.↑ | S.U.↓ | D.U.↓ | R.D.↑ | R.A.↑ | W.F.↑ |
| $o^3$ | 0.0771 | 0.0627 | 0.9675 | 0.9330 | 0.8960 | 0.1939 | 0.1255 | 0.9677 | 0.9330 | 0.8960 | 0.1843 | 0.5189 | 0.9675 | 0.9330 | 0.8960 |
| Ours | **0.0425** | **0.0365** | **0.9675** | **0.9330** | **0.9083** | **0.1333** | **0.1044** | **0.9683** | **0.9330** | **0.9083** | **0.1322** | **0.1062** | **0.9683** | **0.9330** | **0.9083** |

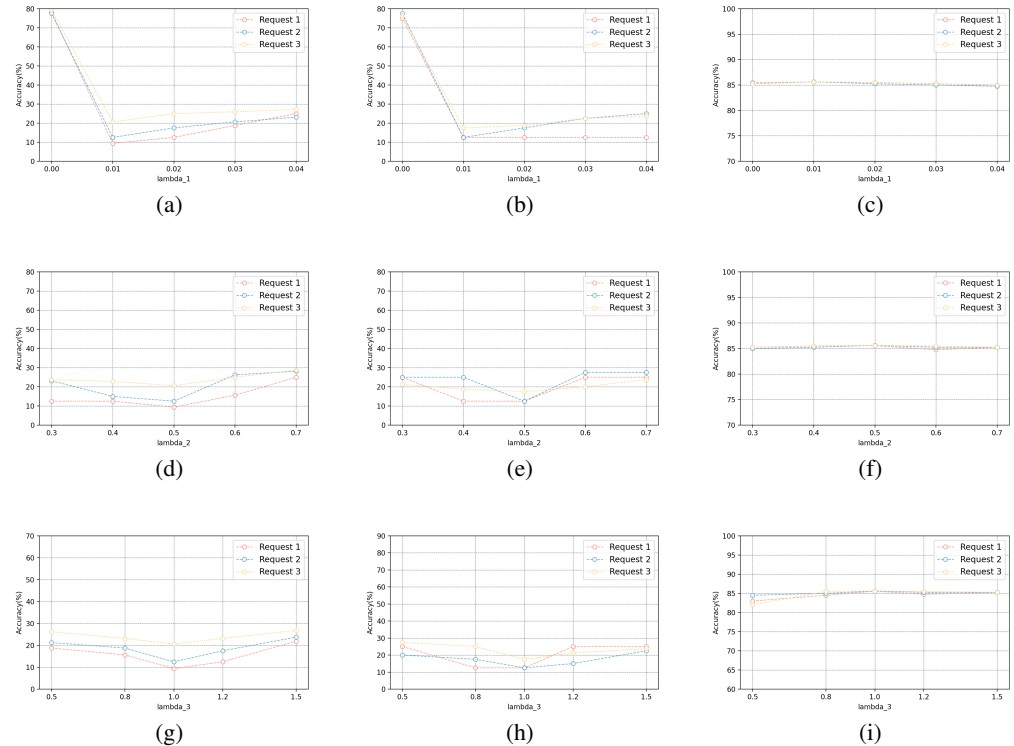

Figure 8: Experimental results on the TOFU dataset. The left column shows the results of S.U., the middle column shows the results of D.U., and the right column shows the results of R.D.. (a) (b) (c)The result of the hyperparameter $\lambda_1$. (d) (e) (f)The result of the hyperparameter $\lambda_2$(S.U.). (g) (h) (i)The result of the hyperparameter $\lambda_3$(S.U.).

