# OpenReview forum: "Rotation Control Unlearning: Quantifying and Controlling Continuous Unlearning for LLM with The Cognitive Rotation Space"
_ICLR.cc/2026/Conference — ICLR 2026 Conference Withdrawn Submission_

### Official Review · Reviewer_UTeN · 2025-10-29

**Soundness:** 3
**Presentation:** 3
**Contribution:** 2
**Rating:** 6
**Confidence:** 5

**Summary:**

The paper proposes Rotation Control Unlearning (RCU), a novel approach for continual machine unlearning in Large Language Models that addresses two major challenges: Cumulative utility degradation across multiple unlearning requests and a Lack of precise control and quantification over the unlearning process. RCU reinterprets LoRA-based updates as rotations in a cognitive representation space. The method introduces:  1) a skew-symmetric loss to model LoRA updates as rotational transformations, 2) an orthogonal rotation axis loss to ensure perpendicular update directions across sequential unlearning requests, thereby minimizing interference and catastrophic forgetting, and 3) a distributional shift compensator that produces rotational salience weights, enabling precise auxiliary quantification of unlearning effects. RCU is tested on ScienceQA and TOFU benchmarks, achieving effective continual unlearning without relying on retained datasets.

**Strengths:**

1. By modeling unlearning as rotations in a latent space, RCU introduces a mathematically grounded and interpretable framework for tracking and controlling knowledge removal.

2. Experiments on TOFU and ScienceQA demonstrate RCU’s unlearning efficacy and utility preservation.

3. The introduction of rotational angles and salience weights provides a precise, tunable metric for unlearning strength.

**Weaknesses:**

1. The training procedure involves a combination of losses for LoRA and OOD detection, making the optimization process relatively complex. However, the paper does not provide sufficient details on the choice of loss weights (Lambda values). A sensitivity analysis should be conducted to demonstrate the model's robustness to different hyperparameter settings and justify the selected values.

2. The paper would benefit from improved writing quality. The rationale for using rotation angle is not clear, and there are several typos and inconsistencies, for example, in Table 4, the word "physics" appears inappropriately or misspelled.

**Questions:**

Could the authors provide justification for the choice of the lambda hyperparameters used in the loss functions? Given that the training objective combines multiple loss terms (e.g., for LoRA tuning and OOD detection), the values of these weights significantly affect model behavior. A robustness analysis—such as an ablation or sensitivity study—should be conducted to evaluate how performance varies with different lambda settings across tasks or datasets.

---

> ### Author Response · Authors · 2025-11-23
> **We sincerely appreciate the reviewer's insightful suggestions and will respond to each comment and question earnestly.**
>
> **Q1: The training procedure involves a combination of losses for LoRA and OOD detection, making the optimization process relatively complex. However, the paper does not provide sufficient details on the choice of loss weights (Lambda values). A sensitivity analysis should be conducted to demonstrate the model's robustness to different hyperparameter settings and justify the selected values.**
>
> **A1**: We thank the reviewer for their valuable feedback. We fully agree that the selection of hyperparameters in the loss function is crucial for model performance. To address the lack of sensitivity analysis in the original manuscript, we have supplemented a detailed sensitivity analysis of lambda in Appendix A.7. Experimental results demonstrate that our model exhibits strong robustness under the current hyperparameter configuration, validating that the chosen lambda value is appropriate.
>
> **Q2: The paper would benefit from improved writing quality. The rationale for using rotation angle is not clear, and there are several typos and inconsistencies, for example, in Table 4, the word "physics" appears inappropriately or misspelled.**
>
> **A2**: Thank you for your valuable suggestions.
>
> We have significantly revised our article, further emphasizing the relationship between our core issue and the proposed method in the introduction section and section 3.1 of the Methodology. Additionally, in section 3.1, we provided further discussion on the reasons for using the rotation angle.
>
> Regarding the reasons for using the rotation angle, we offer the following explanation:
>
> In the Methodology section, we redefine parameter updates as rotations within the original parameter space. Since rotation is a rigid transformation **[1]**, the update of BA only affects the change in the rotation angle. Therefore, the unlearning process is directly driven by this change in the angle, allowing us to precisely quantify the degree of unlearning in the LLM using the rotation angle.
>
> **[1] Mortari D. On the rigid rotation concept in n-dimensional spaces[J]. The Journal of the astronautical sciences, 2001, 49(3): 401-420.**

---

### Official Review · Reviewer_JKXu · 2025-10-31

**Soundness:** 4
**Presentation:** 4
**Contribution:** 4
**Rating:** 6
**Confidence:** 4

**Summary:**

The paper proposes Rotation Control Unlearning (RCU), a novel method for Large Language Model (LLM) unlearning that precisely quantifies and controls the unlearning degree during continuous unlearning requests.
They propose the RCU method, which reformulates the LoRA update process as rotations within a "cognitive rotation space". This approach allows RCU to effectively unlearn information continuously without needing a retained dataset.
The method introduces 3 new key components:
1. Skew Symmetric Loss: A loss function designed to construct the cognitive rotation space. This constraint ensures the LoRA parameter updates behave like rotations.
2. Rotational Salience Weight: A metric derived from an OOD detector that is used to precisely quantify and control the degree of unlearning (i.e., the rotation angle) for any given request.
3. Orthogonal Rotation Axes Regularization: A loss function that enforces mutually perpendicular rotation directions for consecutive unlearning requests. This minimizes interference between different unlearning tasks and directly addresses the problem of cumulative catastrophic utility loss.

**Strengths:**

**1. Fine-grained Control over Unlearning Intensity:**

The paper introduces a novel perspective by modeling the unlearning strength as a controllable variable through rotational scaling. This design allows the model to precisely adjust the degree of forgetting at a fine-grained level. Such a mechanism provides more flexibility and interpretability for selective and controlled unlearning.

**2. Theoretical Novelty through the Cognitive Rotation Space:**

The proposed Cognitive Rotation Space offers a strong theoretical contribution. By formulating unlearning as a spectral-space rotation governed by skew-symmetric transformations, the authors connect geometric orthogonality with cognitive forgetting behavior. This framework is both conceptually elegant and mathematically grounded, bringing a fresh theoretical lens to the study of machine unlearning.

**3. Strong Empirical Performance and Robustness:**

The experimental results are impressive; the method demonstrates consistently strong unlearning performance across two benchmarks while significantly alleviating cumulative catastrophic utility loss in continual unlearning settings. The comprehensive evaluation convincingly shows that the proposed approach achieves an excellent balance between forgetting precision and retention stability.

**Weaknesses:**

**1. Limited Orthogonality Scope:**

The current design of the orthogonal rotation loss \mathcal{L}_o appears to only enforce pairwise orthogonality between the current and the immediately preceding rotation axes. However, it does not ensure global orthogonality across all historical rotations. As the number of unlearning requests T grows, earlier forgetting directions may be reprojected onto new ones, potentially leading to knowledge leakage or cumulative utility degradation. Have the authors considered enforcing global orthogonality, and is it necessary for maintaining long-term stability in continual unlearning?

**2. Granularity of the Rotational Salience Weight $\beta$:**

In the RCU algorithm, is the $\beta$ obtained through the OOD detection and Distributional Shift Compensator shared among all forget data within one unlearning process, or is a separate $\beta$ computed individually for each data sample?

**3.Computational Overhead and Efficiency Concerns:**

Since the proposed approach relies on OOD detection to obtain a sample-dependent $\beta$ value, it seems that an additional forward pass through the OOD module may be required for each input during inference. Will this introduce noticeable latency? Moreover, during training, the optimization process involves multiple components and loss terms, which might increase computational complexity and training time. A more detailed discussion or empirical analysis of the computational overhead would strengthen the paper.

**4. Results Not Consistent with Prior Baselines.**

After reviewing the O³ paper, I notice that the two papers report identical numbers for the other baselines, but this paper’s results are worse than those reported in O³ specifically for the O³ method itself. Given that the remaining unlearning methods appear unchanged across both papers, could the authors clarify the source of this discrepancy?

**Questions:**

Questions are included in the weakness section.

---

> ### Author Response · Authors · 2025-11-23
> **We sincerely appreciate the reviewer's insightful suggestions and will respond to each comment and question earnestly.**
>
> **Q1: The current design of the orthogonal rotation loss \mathcal{L}_o appears to only enforce pairwise orthogonality between the current and the immediately preceding rotation axes. However, it does not ensure global orthogonality across all historical rotations. As the number of unlearning requests T grows, earlier forgetting directions may be reprojected onto new ones, potentially leading to knowledge leakage or cumulative utility degradation. Have the authors considered enforcing global orthogonality, and is it necessary for maintaining long-term stability in continual unlearning?**
>
> **A1**: The use of relative rotation space is based on a comprehensive consideration of method design and training efficiency. Thanks to the characteristics of the high-dimensional geometric space, the randomly generated vectors naturally tend to be orthogonal in this space, which enables adjacent rotation axes to automatically approach a vertical relationship. Therefore, the relative rotation mechanism can achieve an effect similar to that of the enforcing global orthogonality with almost no additional optimization cost.
>
> Furthermore, directly using enforcing global orthogonality incurs a significant amount of computational cost pressure. At the third unlearning request, its single-time computational cost is 12.88 GFLOPs. In contrast, the computational cost of relative rotation is 4.29 GFLOPs. Moreover, as the number of unlearning requests increases, the computational cost brought about by directly using enforcing global orthogonality increases geometrically.
>
> **Q2: In the RCU algorithm, is the Beta obtained through the OOD detection and Distributional Shift Compensator shared among all forget data within one unlearning process, or is a separate Beta computed individually for each data sample?**
>
> **A2**: In the RCU algorithm, the Beta value is not shared across all data throughout the entire unlearning process, nor is it computed individually for each data sample. Instead, it is calculated separately for each batch. The specific procedure is as follows:
>
> (1)	First, the unlearning dataset is divided into multiple batches.
>
> (2)	Then, for each batch, prompts are generated for every sample and tokenization is performed.
>
> (3)	Next, for each batch, all OOD model components are traversed, and the maximum OOD score from all samples in that batch is selected as the batch's weight. This weight is then fed into the Distributional Shift Compensator to derive the final Beta value for that batch.
>
> As a result, each batch has an independent Beta value, which is applied to all samples within that batch, achieving batch-level sharing.
>
> **Q3: Since the proposed approach relies on OOD detection to obtain a sample-dependent Beta value, it seems that an additional forward pass through the OOD module may be required for each input during inference. Will this introduce noticeable latency? Moreover, during training, the optimization process involves multiple components and loss terms, which might increase computational complexity and training time. A more detailed discussion or empirical analysis of the computational overhead would strengthen the paper.**
>
> **A3**: The computation of the OOD module introduces a certain degree of latency, which is unavoidable.
>
> For question about computational:
>
>  During the training process, the computational overhead incurred by multiple loss functions and our proposed novel LoRA update paradigm amounts to 35.31 TFLOPs. During the inference phase, the overall computational overhead of our method is 59.064 GFLOPs. Furthermore, the storage cost of our method is reduced to 16 MB, compared to 39 MB in O3. We have included the discussion on computational overhead in Appendix A.8.
>
> **Q4: After reviewing the O³ paper, I notice that the two papers report identical numbers for the other baselines, but this paper’s results are worse than those reported in O³ specifically for the O³ method itself. Given that the remaining unlearning methods appear unchanged across both papers, could the authors clarify the source of this discrepancy?**
>
> **A4**: Thank you for your question. Our results in the original text are based on the O^3 code. We ran three different results with different random seeds using the O^3 code. However, the results of O^3 are relatively unstable and are also affected by the limited computing resources we have. So we have updated the O^3 results and some other results in the new version. And for the reproduced results, we marked them with * in the new version.
>
> We conducted multiple experiments in our 2 A6000 GPUs and the average results have been updated as follows: S.U.: 15.659, 22.49, 26.66; D.U.: 14.67, 20.17, 23.56; R.D.: 85.25, 85.5, 85.25; R.A.: 89, 89, 89; W.F.: 86.3, 86.3, 86.3. After the update, the performance of our RCU method still significantly outperforms that of the O^3 method.

---

### Official Review · Reviewer_xkUj · 2025-11-01

**Soundness:** 2
**Presentation:** 2
**Contribution:** 2
**Rating:** 2
**Confidence:** 4

**Summary:**

This paper addresses the problem of machine unlearning in LLMs. The authors noted that existing methods rely on retained dataset to preserve utility, and become impractical for larger and more practical settings, and could lead to catastrophic utility loss in continuous setting.The authors propose rotation control unlearning (RCU) that interprets LoRA updates as rotations within a specific cognitive rotation space. The authors claim that rotation angle corresponds to the degree of forgetting. The authors also introduce orthogonal rotation axes loss to allow multiple unlearning steps. The authors conduct experiments on a QA and TOFU benchmark on a single Llama non-instruct model.

**Strengths:**

The studied problem is important and the proposed method shows its effectiveness to some extent.

**Weaknesses:**

Overall, there are several weaknesses the authors should address. Most importantly, why can we quantify the degree of unlearning using rotation angle? This paper does not provide conceptual, empirical, or mathematical justification for why unlearning itself should correspond to a rotation angle in rotation space. Also, the writing needs significant improvement. Many times throughout the paper, the authors use a term extensively without explanations (for example, the cognitive rotation space, what is that?) The authors should explain some of these terms in the preliminary section. The current preliminary section is essentially a ‘related work’ section. The method section also reads as a list of techniques without explaining why they are needed or how they relate to the core problem of unlearning. Overall, the paper does not convincingly argue that RCU is an effective or principled approach to unlearning.

**Questions:**

I also have concerns regarding experiments. Moving into 2026, this setting feels too limited to demonstrate practical relevance. The authors should try larger models on way more unlearning requests to prove effectiveness of their methods.Also, the evaluation metrics are not well justified and not properly defined. For the TOFU task, why not use the eval metrics provided by the original authors? For example, the p-value statistical test and the average of Rouge, Answer Probability, and Truth Ratio? The authors should provide more contexts about these evaluations and justify their choices.

While [1] is a recent work, [1] has achieved ideal unlearning by masking out the training signal of TOFU in their corpus. According to [1] results, an ideally unlearned model will achieve ideal forget quality vs utility trade-off. Also, forgetting ROUGE isn’t always better. This translates to the authors choice of SU and DU, where the authors are claiming that lower is always better. I strongly recommend the authors to use TOFU’s eval metrics and compare their results with LMLM from [1].  Also, the authors should compare computational costs, since we do want a simple and efficient algorithm for unlearning.

Lastly,  [2] has pointed out that unlearning isn’t always robust. More experiments on the robustness of their methods should be presented in the paper.

[1] Zhao, L., Zalouk, S., Belardi, C. K., Lovelace, J., Zhou, J. P., Weinberger, K. Q., ... & Sun, J. J. (2025). Pre-training Large Memory Language Models with Internal and External Knowledge. arXiv preprint arXiv:2505.15962.
[2] Łucki, J., Wei, B., Huang, Y., Henderson, P., Tramèr, F., & Rando, J. (2024). An adversarial perspective on machine unlearning for ai safety. arXiv preprint arXiv:2409.18025.

---

> ### Author Response · Authors · 2025-11-23
> **We sincerely appreciate the reviewer's insightful suggestions and will respond to each comment and question earnestly.**
>
> **Q1: Most importantly, why can we quantify the degree of unlearning using rotation angle? This paper does not provide conceptual, empirical, or mathematical justification for why unlearning itself should correspond to a rotation angle in rotation space.**
>
> **A1**: Thank you for your question. Our approach redefines the parameter updates in LLMs as rotational transformations within the original parameter space. Under this framework, the update of the BA matrix is equivalent to driving the rotation of the entire parameters. Since rotation is the rigid transformation **[1]**, the original parameter norms remain largely constant.
>
>  As a result, the essence of the unlearning process is captured by the change in the rotation angle. This allows us to directly use the rotation angle to quantify the degree of unlearning. This relationship has been experimentally validated in Fig. 2(a)(b), where Beta is proportional to the rotation angle, as proven in Theorem 2. We have also updated the relevant content in the methodology section.
>
> **[1] Mortari D. On the rigid rotation concept in n-dimensional spaces[J]. The Journal of the astronautical sciences, 2001, 49(3): 401-420.**
>
> **Q2: The cognitive rotation space, what is that? The authors should explain some of these terms in the preliminary section. The current preliminary section is essentially a ‘related work’ section. The method section also reads as a list of techniques without explaining why they are needed or how they relate to the core problem of unlearning. Overall, the paper does not convincingly argue that RCU is an effective or principled approach to unlearning.**
>
> **A2**: Thank you for your question. We have updated our description regarding the cognitive rotation space. The cognitive rotation space is clearly defined in the methodology section as an ideal rotation space approximation based on I+BA (proof part in Section 3.1), which originates from the theory of Lie groups.
>
>   Furthermore, we have significantly updated our paper. In the introduction section, we highlight the core issues we need to address, while in the methodology section, we elaborate on the relationship between our method and the core issues and explain the reasons for using the rotation technique.
>
> **Q3: I also have concerns regarding experiments. Moving into 2026, this setting feels too limited to demonstrate practical relevance. The authors should try larger models on way more unlearning requests to prove effectiveness of their methods. Also, the evaluation metrics are not well justified and not properly defined. For the TOFU task, why not use the eval metrics provided by the original authors? For example, the p-value statistical test and the average of Rouge, Answer Probability, and Truth Ratio? The authors should provide more contexts about these evaluations and justify their choices.**
>
> **A3**: For questions about larger models and more unlearning requests:
>
> (1) we conducted experiments on Llama2-7B, whose size is sufficient to represent the performance of the methods related to continual unlearning. The O^3 **[1]** method also used Llama2-7B for experiments and has received widespread recognition.
>
> (2) The current unlearning request actually depends on the size of the dataset. Excessive segmentation of the dataset may fail to reflect the balance between unlearning and retaining knowledge during continual unlearning. Our current unlearning requests have been designed with this consideration in mind during experimentation. O^3 **[1]** also maintains such an experimental setup.
>
> For questions about evaluation metrics , we follow the evaluation paradigm of the O^3 framework **[1]**, which is specifically designed to assess unlearning effects in continuous learning environments (e.g., S.U., D.U., R.D., and U2R). Its metrics effectively measure the balance of unlearning and utility loss, and have gained significant recognition. We have added the comparison results of Truth Ratio and ROUGE-L in Appendix A.9. Our methods all achieved the best results.
>
> **[1]Gao C, Wang L, Ding K, et al. On large language model continual unlearning. In The Thirteenth International Conference on Learning Representations, 2025.**

---

> ### Author Response · Authors · 2025-11-23
> **other problems**
>
> **Q4: While LMLM is a recent work, LMLM has achieved ideal unlearning by masking out the training signal of TOFU in their corpus. According to LMLM results, an ideally unlearned model will achieve ideal forget quality vs utility trade-off. Also, forgetting ROUGE isn’t always better. This translates to the authors choice of SU and DU, where the authors are claiming that lower is always better. I strongly recommend the authors to use TOFU’s eval metrics and compare their results with LMLM. Also, the authors should compare computational costs, since we do want a simple and efficient algorithm for unlearning.**
>
> **A4**:
> For the question about LMLM:
>
> (1) we have thoroughly read about the LMLM method **[1]**, which achieves parameter-efficient, high-fact-accuracy, verifiable knowledge updates, and instant unlearning by externalizing factual knowledge into an editable database. However, it should be noted that LMLM and our method are fundamentally different in their approaches. LMLM primarily focuses on external data, replacing direct factual recall with retrieval-augmented generation to produce required responses. In contrast, our method, along with the comparative methods we examined, emphasizes achieving unlearning through modifications to the internal parameter space.
>
> (2) Furthermore, in the context of continual unlearning, our experiments involve multiple unlearning trials on Llama2-7B. Comparing this with the computational overhead introduced by LMLM （whose original authors primarily conducted independent experiments on models of smaller scales such as Llama2-176M and GPT2-335M）would be inherently unfair. A separate discussion of these methods is more aligned with academic rigor.
>
> (3) Finally, both LMLM and our method represent valuable explorations in the field of unlearning. LMLM prioritizes rapid effectiveness by integrating ideas akin to data distillation **[2]** and RAG **[3]**, thereby achieving stronger immediacy and applicability. Our method, tailored for the continuous unlearning scenario, focuses on balancing the unlearning and model utility across multiple requests while addressing cumulative catastrophic utility loss **[4]**. So we cite the LMLM in the preliminary.
>
> For the question eval metrics:
>
> To further illustrate the balance between forgetting and utility preservation that our method guarantees, we introduced the indicator U^2R **[4]** in Figure 2(c) as a more intuitive evaluation criterion. Furthermore, we have also provided the comparison results of the evaluation indicators for TOFU in Appendix A.9.
>
> **[1] Zhao, L., Zalouk, S., Belardi, C. K., Lovelace, J., Zhou, J. P., Weinberger, K. Q., ... & Sun, J. J. (2025). Pre-training Large Memory Language Models with Internal and External Knowledge. arXiv preprint arXiv:2505.15962.**
>
> **[2] Sun P, Shi B, Yu D, et al. On the diversity and realism of distilled dataset: An efficient dataset distillation paradigm[C]//Proceedings of the IEEE/CVF Conference on Computer Vision and Pattern Recognition. 2024: 9390-9399.**
>
> **[3] Arslan M, Ghanem H, Munawar S, et al. A Survey on RAG with LLMs[J]. Procedia computer science, 2024, 246: 3781-3790.**
>
> **[4] Gao C, Wang L, Ding K, et al. On large language model continual unlearning. In The Thirteenth International Conference on Learning Representations, 2025.**
>
> **Q5: Lastly, unlearning isn’t always robust. More experiments on the robustness of their methods should be presented in the paper.**
>
> **A5**: Thank you for raising this important question. We have incorporated U^2R [1], Truth Ratio and ROUGE-L as the criteria for evaluating the performance of our method, in order to verify the effectiveness of our approach.
>
> **[1] Gao C, Wang L, Ding K, et al. On large language model continual unlearning. In The Thirteenth International Conference on Learning Representations, 2025.**

---

### Official Review · Reviewer_s53n · 2025-11-04

**Soundness:** 2
**Presentation:** 3
**Contribution:** 2
**Rating:** 2
**Confidence:** 4

**Summary:**

The paper improves on the continual framework of unlearning named $O^3$ [1], which consists of orthogonal LoRA adapters for different unlearning requests, a OOD detector trained with a novel contrastive entropy loss and utilizes a global-aware scoring mechanism, and a soft inference algorithm to detect unlearned data during inference.

The paper introduces rotation matrices for updating the LoRA adaptors and constrain the rotation angles to be orthogonal for every unlearning request. They update the consequent OOD detector to take this modification into account.


[1] Gao, Chongyang, et al. "On large language model continual unlearning." ICLR 2025

**Strengths:**

- The paper is well-written for the most part.
- The empirical results look strong, even though its not completely convincing.

**Weaknesses:**

- Please improve the figures. They are hard to see without a lot of zooming in.
- The metrics in Table 1 for $O^3$ do not match with the original paper. In this case, please include multiple runs to demonstrate the improvement properly. This is a major weakness of this paper because of how closely it follows $O^3$ and claims improvement.
- Please include the $U^2R$ metric from the $O^3$ paper to demonstrate improved utility preservation over  $O^3$ .
- From Equation 4, 5 : $C = BA$ and $R = exp(C) = I + BA$.  This seems very strange. Is there any evidence why this approximation may hold , can you please provide evidence that $C << I$ from your experiments ?
- The above point makes it very hard to understand whether the method legitimately works due to the rotation aspect or the performance is still an artifact of $O^3$ due to the high similarity of constraints and other frameworks.

**Questions:**

- Should Eq. 3 be $W = W + BA$ ?
- One of the central claims of the paper is that the idea of updating LoRA parameters using rotation matrices will improve the catastrophic utility degradation. Why is it so ? Why is this better than $O^3$? Is there any intuition ?
- In Eq. 11, why is the use of relative rotation space needed ? Can we directly use $R_t$ and $R_{t-1}$ ?


Please address the Weaknesses and the above questions.\
I am willing to raise my score based on the clarification.

---

> ### Author Response · Authors · 2025-11-23
> **We sincerely appreciate the reviewer's insightful suggestions and will respond to each comment and question earnestly.**
>
> **Firstly，We sincerely thank you for highlighting the writing issues in our manuscript and have carefully incorporated all suggested improvements in the new version.**
>
> **Q1: The metrics in Table 1 for O^3 do not match with the original paper. In this case, please include multiple runs to demonstrate the improvement properly. This is a major weakness of this paper because of how closely it follows O^3 and claims improvement.**
>
> **A1**: We noticed that the O^3 indicator in Table 1 differs from the data in the original paper. We conducted multiple experiments in our 2 A6000 GPUs and the average results have been updated as follows: S.U.: 15.659, 22.49, 26.66; D.U.: 14.67, 20.17, 23.56; R.D.: 85.25, 85.5, 85.25; R.A.: 89, 89, 89; W.F.: 86.3, 86.3, 86.3. After the update, the performance of our RCU method is still the best compared to the O^3 method. Additionally, we updated the results of other several baseline methods. And for the reproduced results, we marked them with * in the new version.
>
> **Q2: Please include the U^2 R metric from the O^3 paper to demonstrate improved utility preservation over.**
>
> **A2**: We have added the results of U^2R. The detailed comparison results can be seen in Figure 2 (c).
>
> **Q3: From Equation 4, 5 : C=BA and R=exp(C)=I+BA. This seems very strange. Is there any evidence why this approximation may hold , can you please provide evidence that C<<I from your experiments.**
>
> **A3**: In Table 4 of Appendix A.6, we present the magnitude results of LoRA parameters, and for more intuitive understanding, we visualize the changes in parameter magnitudes during LoRA updates in Figure 6 (a)(b), which respectively display the value variations of all lora_A and lora_B parameters throughout the global update process in the trainable layer. Here, we observe that the magnitude of lora_B parameters is 1e-5, while that of lora_A parameters is 1e-5, thereby validating that the size of C=BA is substantially smaller than I.
>
> **Q4: The above point makes it very hard to understand whether the method legitimately works due to the rotation aspect or the performance is still an artifact of O^3 due to the high similarity of constraints and other frameworks.**
>
> **A4**: The effectiveness of our method does not depend on O^3; instead, it is based on solid theoretical and practical grounds.
>
> （1） From a theoretical perspective, we restructured the unlearning process by introducing the theory of Lie groups, converting the parameter update into precise rotational transformations in the original space, and representing the parameter shift as changes in the rotation angle. As a conformal transformation, rotation helps maintain the unlearning ability of historical unlearning requests. Compared to the uncontrollable parameter update mechanism in O3, our method can more effectively alleviate the cumulative catastrophic utility loss caused by continual unlearning.
>
> （2）From the perspective of interpretability, our method has excellent interpretability. The experimental results in Figure 2(a)(b) clearly demonstrate the direct correlation between the rotation angle and the unlearning effect (Beta is proportional to the size of the rotation angle, as proved in Theorem 1), visually presenting the impact of different rotation angles on the unlearning process. However, the O^3 method relies more on experience for the setting of LoRA weights and lacks an interpretable mechanism for support.
>
> （3）From the perspective of loss design, the L_sk loss we proposed is used to constrain the rotational transformation in the parameter space and works in conjunction with our LoRA update mechanism; the L_o loss, on the other hand, ensures that the rotation axes of adjacent tasks are perpendicular to each other by forcing them to be so, thereby preventing interference in the direction of parameter updates and maintaining stable unlearning ability across multiple unlearning tasks. In contrast, O3 only constrains the adjacent LoRA_A parameters to be perpendicular to reduce the influence between tasks. This constraint may be weakened due to the participation of LoRA_B, and its constraint ability gradually declines as the number of tasks increases.
>
> In summary, our approach has a solid theoretical foundation and practical mechanism design. Its effectiveness does not rely on the O^3 method, but stems from its inherent rotation mechanism and loss strategy.

---

> ### Author Response · Authors · 2025-11-23
> **other problems**
>
> **Q5: One of the central claims of the paper is that the idea of updating LoRA parameters using rotation matrices will improve the catastrophic utility degradation. Why is it so ? Why is this better than O^3? Is there any intuition ?**
>
> **A5**: The rotation mechanism can effectively alleviate catastrophic utility loss. The key lies in its ability to act as the rigid transformation **[1]**, maintaining its norm while updating the parameter direction. This ensures the stability of the knowledge structure. The continuous unlearning process is transformed into an asymptotic rotation in the parameter space: each new request triggers only one rotation, and this operation does not change the relative relationship of parameters defined by previous rotations, thereby avoiding the decline of existing unlearning effects.
>
>  In contrast, the O^3 method relies on orthogonal constraints (making LoRA_A parameters perpendicular), which have inherent limitations: on one hand, the introduction of LoRA_B will interfere with the effectiveness of this constraint; on the other hand, as the cumulative unlearning requests increase, the effectiveness of this orthogonal constraint will continuously decline, making it difficult for it to cope with the challenge of continual unlearning.
>
> **[1] Mortari D. On the rigid rotation concept in n-dimensional spaces[J]. The Journal of the astronautical sciences, 2001, 49(3): 401-420.**
>
> **Q6: In Eq. 11, why is the use of relative rotation space needed ? Can we directly use R_t and R_(t-1) ?**
>
> **A6**: The use of relative rotation space is based on a comprehensive consideration of method design and training efficiency. Our method transforms the parameter updates of the language model into changes in rotation angles, thereby converting the parameter interference problem between adjacent unlearning tasks into a constraint problem between rotation angles. Thanks to the characteristics of the high-dimensional geometric space, the randomly generated vectors naturally tend to be orthogonal in this space, which enables adjacent rotation axes to automatically approach a vertical relationship. Therefore, the relative rotation mechanism can achieve an effect similar to that of the global rotation space with almost no additional optimization cost.
>
> Furthermore, directly using R_t and R_{t-1} for global vertical processing incurs a significant amount of computational cost pressure. At the third unlearning request, its single-time computational cost is 12.88 GFLOPs. In contrast, the computational cost of relative rotation is 4.29 GFLOPs. Moreover, as the number of unlearning requests increases, the computational cost brought about by directly using R_t and R_{t-1} increases geometrically.

---

### Note · Authors · 2025-12-04

I have read and agree with the venue's withdrawal policy on behalf of myself and my co-authors.